# Ultra-High Repetition Rate Terahertz Time-Domain Spectroscopy for Micrometer Layer Thickness Measurement

**DOI:** 10.3390/s21165389

**Published:** 2021-08-10

**Authors:** Kevin Kolpatzeck, Xuan Liu, Lars Häring, Jan C. Balzer, Andreas Czylwik

**Affiliations:** Chair of Communication Systems (NTS), Faculty of Engineering, University of Duisburg-Essen (UDE), 47057 Duisburg, Germany; xuan.liu@uni-due.de (X.L.); haering@nts.uni-duisburg-essen.de (L.H.); jan.balzer@uni-duisburg-essen.de (J.C.B.); czylwik@nts.uni-duisburg-essen.de (A.C.)

**Keywords:** terahertz, time-domain spectroscopy, mode-locked laser diode, synchronization, interferometer

## Abstract

Terahertz time-domain spectroscopy systems driven by monolithic mode-locked laser diodes (MLLDs) exhibit bandwidths exceeding 1 THz and a peak dynamic range that can compete with other state-of-the-art systems. Their main difference compared to fiber-laser-driven systems is their ultra-high repetition rate of typically dozens of GHz. This makes them interesting for applications where the length of the terahertz path may not be precisely known and it enables the use of a very short and potentially fast optical delay unit. However, the phase accuracy of the system is limited by the accuracy with which the delay axes of subsequent measurements are synchronized. In this work, we utilize an all-fiber approach that uses the optical signal from the MLLD in a Mach–Zehnder interferometer to generate a reference signal that we use to synchronize the detected terahertz signals. We demonstrate transmission-mode thickness measurements of stacked layers of 17μm thick low-density polyethylene (LDPE) films.

## 1. Introduction

Since its inception in the late 1980s, terahertz time-domain spectroscopy (THz-TDS) has proven its potential for a large variety of applications from material characterization and imaging to fundamental research [1,2,3,4,5]. Fiber-coupled systems using compact photoconductive emitters and detectors have reached a high level of maturity and can nowadays be found in hundreds of terahertz labs worldwide. Besides conventional THz-TDS systems driven by a mode-locked fiber laser and using a free-space optical delay unit (ODU) for sampling the received terahertz field, different approaches that, among other things, improve the dynamic range, measurement speed, system size and robustness, as well as system cost have been demonstrated and commercialized. For example, several groups have proposed methods to eliminate the need for a mechanical ODU. These include asynchronous optical sampling (ASOPS) [6,7,8], electrically controlled optical sampling (ECOPS) [9], optical sampling by cavity tuning (OSCAT) [10,11], and single-laser polarization-controlled optical sampling (SLAPCOPS) [12].

Besides these developments, there have been efforts to reduce the system complexity and cost by finding alternative light sources for the spectrometer. Early works have proposed multi-mode laser diodes in a so-called cross-correlation spectroscopy (CCS) or quasi-time-domain spectroscopy (QTDS) approach [13,14,15]. Recently, monolithic mode-locked laser diodes (MLLDs) have gained attraction as light sources for THz-TDS [16,17]. They are compact, robust, electrically pumped, and provide ample power at telecom wavelengths without amplification. While the operating principle of MLLD-driven systems is similar to that of fiber laser-driven systems, their repetition rate is orders of magnitude higher, typically in the dozens of GHz. Because of this fundamental property, we call this approach ultra-high repetition rate (UHRR) THz-TDS [18]. Like the conventional THz-TDS systems, it requires an ODU to sample the received terahertz field. However, because identical terahertz and infrared pulses arrive at the terahertz receiver with the repetition rate of the MLLD, the delay range of the ODU needs to cover only one repetition period of the MLLD. This short delay range enables the use of very compact and fast ODUs. That makes voice coil-driven ODUs particularly attractive. However, the bandwidth and accuracy of THz-TDS systems depend strongly on the accuracy of the delay axis [19,20].

The effects of positioning errors of the ODU have been subject to extensive investigations [21,22,23,24]. The simplest type of error is a random delay offset between successive measurements. Because the optical delay cannot be distinguished from the delay of the received terahertz signal, this type of error directly affects the system’s suitability for the characterization of dielectric layers or for high-resolution imaging. There are, in principle, two ways of ensuring an accurate delay axis. One way is to construct the ODU from highly precise mechanical parts and incorporate electronic circuits that accurately monitor the state of the ODU. This approach however makes the ODU extremely expensive. The other way is monitoring the delay axis with a Mach–Zehnder interferometer which includes the ODU in one of its arms. This has been successfully demonstrated by polarization multiplexing the optical signal from a continuous-wave laser through the ODU with a pair of polarization beam splitters [22]. The non-delayed and delayed optical signals from the continuous-wave laser are combined in a photodetector, so that the interference pattern can be observed in the photocurrent. This method has proven to offer a time resolution of 0.53 fs, which is more than sufficient for THz-TDS [22]. Because of the high frequency of the interference pattern, it allows the delay axis to be corrected on a sample-by-sample basis. This method can also be applied to UHRR-THz-TDS. However, the additional laser adds to the system cost and complexity, and polarization multiplexing requires two polarization beam splitters and careful polarization control.

In this work, we demonstrate that the optical signal from the MLLD itself can be used in a Mach–Zehnder interferometer to generate a reference photocurrent in a low-cost photodiode with which we can synchronize the photocurrents from the terahertz receiver. This approach requires minimal additional hardware effort and facilitates accurate processing of the measured terahertz signals. Using this concept, we perform measurements on thin polymer films and demonstrate our UHRR-THz-TDS system’s ability to resolve 17μm thick dielectric layers in transmission mode.

The paper is structured as follows. In Section 2, we first summarize the mathematical background of UHRR-THz-TDS and introduce the system setup with the interferometer. We calculate the interferometer signal and establish its relationship with the detected terahertz signal. In Section 3, we first introduce our experimental setup and demonstrate the signal processing steps. Employing these signal processing steps, we then test the time resolution of the synchronized UHRR-THz-TDS system.

## 2. Theoretical Background

### 2.1. UHRR-THz-TDS Setup without Interferometer

We consider the fiber-coupled THz-TDS system driven by an MLLD depicted in Figure 1. The chirp of the laser pulses is compensated with a section of single-mode fiber. The laser signal is split into a transmit (Tx) and a receive (Rx) path. On the transmit side, a biased antenna-integrated photodiode is used to convert the optical signal from the infrared into the terahertz domain. On the receive side, the optical signal is delayed with an ODU and illuminates a photoconductive detector.

The photocurrent at the output of the photoconductive detector is proportional to the time average of the product of the instantaneous optical power and the terahertz electrical field incident on the photoconductor. We describe the complex optical spectrum of the MLLD as
(1)$$\begin{array}{cc} {\underline{E}}_{\mathrm{MLLD}}\left(\omega \right)=\sum_{k=0}^{N-1}E_{k}\cdot \left\{\delta \left[\omega +\left(\omega_{0}+k\Omega \right)\right]\cdot e^{-j\varphi_{k}}+\delta \left[\omega -\left(\omega_{0}+k\Omega \right)\right]\right. & \left.\cdot e^{j\varphi_{k}}\right\}\;, \end{array}$$
with Ω=2π·F, where *N* denotes the number of laser modes we consider, $E_{k}$ and $\varphi_{k}$ are the amplitude and phase of the *k*-th mode, respectively, $\omega_{0}$ is the angular frequency of the first considered mode, and *F* is the repetition rate of the MLLD. We have shown that the detected photocurrent as a function of the delay τ of the ODU can be calculated as [18]
(2)$$\begin{array}{cc} i_{\mathrm{THz}}\left(\tau \right)\propto 2\cdot \sum_{m=1}^{N-1}\{ & \left|{\underline{H}}_{\mathrm{THz}}\left(m\Omega \right)\right|\cdot \sum_{k=m}^{N-1}\sum_{l=m}^{N-1}E_{k}E_{k-m}E_{l}E_{l-m}\\ \; & \cdot \sin\left[m\Omega \tau +∠H_{\mathrm{THz}}\left(m\Omega \right)+\left(\varphi_{k}-\varphi_{k-m}\right)-\left(\varphi_{l}-\varphi_{l-m}\right)\right]\}\;, \end{array}$$
where ${\underline{H}}_{\mathrm{THz}}\left(m\Omega \right)$ is the value of the complex transfer function of the terahertz system at the angular frequency mΩ. The transfer function ${\underline{H}}_{\mathrm{THz}}\left(m\Omega \right)$ includes the frequency responses of the terahertz transmitter, the terahertz receiver, and the radio channel between the antennas. For the case of perfect chirp compensation, i.e., a linear optical phase, this expression simplifies to
(3)$$\begin{array}{c} i_{\mathrm{THz}}\left(\tau \right)\propto 2\cdot \sum_{m=1}^{N-1}\left\{\left|{\underline{H}}_{\mathrm{THz}}\left(m\Omega \right)\right|\cdot \sum_{k=m}^{N-1}\sin\left[m\Omega \tau +∠{\underline{H}}_{\mathrm{THz}}\left(m\Omega \right)\right]\cdot \sum_{l=m}^{N-1}E_{k}E_{k-m}E_{l}E_{l-m}\right\}\;. \end{array}$$

It describes a signal that is periodic with $T=F^{-1}$ and contains spectral components at the frequencies f=mF,m=1…N−1. The Fourier transform of this expression gives the detected terahertz spectrum. Notably, the phase of each spectral component depends only on the phase of the complex transfer function of the terahertz system. The spectral information of a sample can then be determined, for example, by comparing the spectrum measured with the sample to a reference spectrum measured without the sample.

### 2.2. UHRR-THz-TDS Setup with Interferometer

As mentioned in the introduction, one way of monitoring the delay axis is by using a Mach–Zehnder interferometer which includes the ODU in one of its arms [22]. In this section, we show how the interference of the non-delayed and the delayed optical signal from the MLLD itself can be used to synchronize the delay axes.

Consider the block diagram depicted in Figure 2. A fraction of the optical signals going to the terahertz transmitter respectively receiver is split off with a pair of fiber-optic couplers (couplers 2 and 3) and combined with another fiber-optic coupler (coupler 4) in a pair of photodiodes. The spectra at the two inputs of coupler 4 are
(4)$$\begin{array}{cc} {\underline{E}}_{\mathrm{top}}\left(\omega \right) & \propto \sum_{k=0}^{N-1}E_{k}\cdot \left\{\delta \left[\omega +\left(\omega_{0}+k\Omega \right)\right]\cdot e^{-j\left(\frac{\pi }{2}+\varphi_{k}\right)}+\delta \left[\omega -\left(\omega_{0}+k\Omega \right)\right]\cdot e^{j\left(\frac{\pi }{2}+\varphi_{k}\right)}\right\} \end{array}$$
and
(5)$$\begin{array}{cc} {\underline{E}}_{\mathrm{bottom}}\left(\omega \right) & \propto \sum_{k=0}^{N-1}E_{k}\cdot \{\delta \left[\omega +\left(\omega_{0}+k\Omega \right)\right]\cdot e^{-j\left(\pi +\varphi_{k}\right)}\\ \; & \;\;\;\;\;+\delta \left[\omega -\left(\omega_{0}+k\Omega \right)\right]\cdot e^{j\left(\pi +\varphi_{k}\right)}\}\cdot e^{-j\omega \tau }\\ \; & =\sum_{k=0}^{N-1}E_{k}\cdot \{\delta \left[\omega +\left(\omega_{0}+k\Omega \right)\right]\cdot e^{-j\left[\pi +\varphi_{k}-\left(\omega_{0}+k\Omega \right)\cdot \tau \right]}\\ \; & \;\;\;\;\;+\delta \left[\omega -\left(\omega_{0}+k\Omega \right)\right]\cdot e^{j\left[\pi +\varphi_{k}-\left(\omega_{0}+k\Omega \right)\cdot \tau \right]}\}\;. \end{array}$$

Thus, the spectra at the two outputs of coupler 4 are
(6)$$\begin{array}{cc} {\underline{E}}_{-}\left(\omega \right) & \propto \sum_{k=0}^{N-1}E_{k}\cdot \{\delta \left[\omega +\left(\omega_{0}+k\Omega \right)\right]\cdot e^{-j\varphi_{k}}\cdot \left(e^{-j\frac{\pi }{2}}+e^{-j\left[\frac{3\pi }{2}-\left(\omega_{0}+k\Omega \right)\cdot \tau \right]}\right)\\ \; & \;\;\;\;\;+\delta \left[\omega -\left(\omega_{0}+k\Omega \right)\right]\cdot e^{j\varphi_{k}}\cdot \left(e^{j\frac{\pi }{2}}+e^{j\left[\frac{3\pi }{2}-\left(\omega_{0}+k\Omega \right)\cdot \tau \right]}\right)\} \end{array}$$
and
(7)$$\begin{array}{cc} {\underline{E}}_{+}\left(\omega \right) & \propto \sum_{k=0}^{N-1}E_{k}\cdot \{\delta \left[\omega +\left(\omega_{0}+k\Omega \right)\right]\cdot e^{-j\varphi_{k}}\cdot \left(e^{-j\pi }+e^{-j\left[\pi -(\omega_{0}+k\Omega )\cdot \tau \right]}\right)\\ \; & \;\;\;\;\;+\delta \left[\omega -\left(\omega_{0}+k\Omega \right)\right]\cdot e^{j\varphi_{k}}\cdot \left(e^{j\pi }+e^{j\left[\pi -(\omega_{0}+k\Omega )\cdot \tau \right]}\right)\}\;. \end{array}$$

The photocurrents at the outputs of the two photodiodes are proportional to the time averages of the instantaneous optical powers, i.e., the zero-frequency component of the instantaneous optical power:(8)$$\begin{array}{cc} i_{\mathrm{Interf},-}\left(\tau \right) & \propto F^{-1}\left\{F\left\{F^{-1}\left\{{\underline{E}}_{-}\left(\omega \right)\right\}\left(t\right)\cdot F^{-1}\left\{{\underline{E}}_{-}\left(\omega \right)\right\}\left(t\right)\right\}\left(\omega \right)\cdot H_{\mathrm{LP}}\left(\omega \right)\right\}\\ \; & \propto F^{-1}\left\{\left[{\underline{E}}_{-}\left(\omega \right)*{\underline{E}}_{-}\left(\omega \right)\right]\cdot H_{\mathrm{LP}}\left(\omega \right)\right\}\;, \end{array}$$
where $H_{\mathrm{LP}}\left(\omega \right)$ is the transfer function of an ideal low-pass filter with a cutoff angular frequency that is much lower than Ω. This gives
(9)$$\begin{array}{c} i_{\mathrm{Interf},-}\left(\tau \right)\propto \sum_{k=0}^{N-1}{E_{k}}^{2}\cdot \left\{1-\cos\left[\left(\omega_{0}+k\Omega \right)\cdot \tau \right]\right\} \end{array}$$
and analogously
(10)$$\begin{array}{c} i_{\mathrm{Interf},+}\left(\tau \right)\propto \sum_{k=0}^{N-1}{E_{k}}^{2}\cdot \left\{1+\cos\left[\left(\omega_{0}+k\Omega \right)\cdot \tau \right]\right\}\;. \end{array}$$
The difference between these two photocurrents is
(11)$$\begin{array}{c} i_{\mathrm{Interf}}\left(\tau \right)=i_{\mathrm{Interf},+}\left(\tau \right)-i_{\mathrm{Interf},-}\left(\tau \right)\propto \sum_{k=0}^{N-1}{E_{k}}^{2}\cdot \cos\left[\left(\omega_{0}+k\Omega \right)\cdot \tau \right]\;. \end{array}$$

This expression describes an amplitude-modulated oscillation at the optical carrier frequency in the delay domain. The signal envelope is periodic with *T*. As a side-note, the Fourier transform of this interferogram is the power spectral density of the MLLD. A calculated example is depicted in Figure 3. The interferogram $i_{\mathrm{Interf}}\left(\tau \right)$ exhibits the same periodicity as the detected terahertz photocurrent $i_{\mathrm{THz}}\left(\tau \right)$, but it does not depend on the terahertz spectrometer. Because the interferometer signal does not depend on the terahertz spectrometer, it can serve as a reference between measurements. To reconstruct a synchronized delay axis for the terahertz measurement data, we perform the following steps:We determine the envelope of the interferometer signal.We locate two local maxima of the envelope.We crop the terahertz measurement signal at the locations of the maxima.We assign linearly increasing delay values from 0 to *T* to the sampled terahertz measurement signal in that range.

If we consider a 1550 nm MLLD ($f_{0}\approx 193\;\mathrm{THz}\Leftrightarrow f_{0}^{-1}\approx 5.2\;\mathrm{fs}$), femtosecond-scale synchronization accuracy is possible. Considering a maximum signal frequency of 2 THz of the detected terahertz signal, this accuracy is more than sufficient to maintain the complete phase information between measured terahertz spectra. While in this work only the envelope is used to *synchronize* the delay axis, it should be noted that the carrier-frequency oscillation in the interferogram could also be used for *correcting* a non-linear delay axis similar to the approach shown in [22]. In the following section, we demonstrate a possible implementation of the synchronization procedure.

## 3. Experimental Results and Signal Processing

In this section, we show an experimental realization of the synchronization procedure described in Section 2.2. In Section 3.1, we explain our experimental setup and how the interferogram simplifies our data acquisition. Subsequently, in Section 3.2, we present experimental results and demonstrate how the interferogram enables accurate signal processing for phase-sensitive spectroscopic analysis.

### 3.1. Experimental Setup

A sketch of the experimental setup and a photograph of the measurement setup are depicted in Figure 4 and Figure 5, respectively. We use a Thorlabs FPL1009P Fabry-Perot laser diode as the light source of the terahertz spectrometer. In our previous work, we have found the FPL1009P to be mode-locked [17,18]. The FPL1009P is a convenient choice for this application because of its availability and affordability. The optical signal from the MLLD passes through an isolator to prevent any back-reflections from reaching the laser. We use an 84 m long section of polarization-maintaining (PM) single-mode fiber to compensate the chirp of the MLLD. The signal is split with a 50:50 fused-fiber coupler into the transmit (Tx) and receive (Rx) arms of the spectrometer. In the Rx arm, we use an OZ Optics ODL-650 variable ODU. The delay line has a delay range of 330 ps. We operate it in point-to-point continuous mode. That means we let it run back and forth between the minimum and maximum delay at its maximum speed. A 2 m long section of PM fiber is used in the Tx arm as a “dummy” for the delay line to balance the total fiber lengths in both arms. In both arms, a 90:10 fused-fiber coupler sends 90% of the optical power to the terahertz transmitter respectively receiver and 10% of the optical power to the optical interferometer.

#### 3.1.1. Interferometer

The interferometer consists of a 50:50 fused-fiber coupler that combines the signals from the Tx and Rx arms, and sends them to a pair of photodiodes. The photocurrents from the two photodiodes are transimpedance-amplified and a balanced detection is achieved through a difference amplifier. The voltage difference $v_{\mathrm{Interf}}\left(\tau \right)$ is recorded on channel 1 of a Rohde & Schwarz RTB2004 digital storage oscilloscope (DSO). The acquisition is triggered on the rising-edge of the interferometer signal. We adjust the hold-off time of the trigger so that only one trigger event is captured for each back-and-forth movement of the delay line. We use the high-resolution acquisition mode with a sampling rate of 417 kSa/s.

#### 3.1.2. Terahertz System

The terahertz transmitter is an InGaAs antenna-integrated photodiode-based module (*Toptica #EK-000724*) and the terahertz receiver is an InGaAs photoconductive antenna-based module (*Toptica #EK-000725*). Both modules are from Fraunhofer Heinrich Hertz Institute (HHI). The transmission-mode measurements in this paper are performed with a focused terahertz setup. We use two pairs of plano-convex polymethylpentene (TPX) lenses to focus the terahertz radiation from the transmitter module on the sample, and to focus the transmitted terahertz radiation into the receiver module. The focused spot size in the sample plane is approximately 1 mm. The photocurrent from the receiver is transimpedance-amplified and recorded on channel 2 of the DSO.

### 3.2. Measurement Results and Signal Processing

We demonstrate the feasibility of our synchronization approach by performing transmission measurements on dielectric samples of different thicknesses. For each sample we perform a reference measurement without the sample and a measurement with the sample. In Section 3.2.1 and Section 3.2.2, we show the results and signal processing steps in detail for a 1 mm thick cyclic olefin copolymer (COC) sample. In Section 3.2.3, we then present measurement results for stacked layers of cling film to demonstrate the time resolution of our system.

#### 3.2.1. Synchronization and Cropping in the Time Domain

The signals recorded on the oscilloscope for the case without and with the COC sample are depicted in Figure 6a,b, respectively. The signals from the interferometer $v_{\mathrm{Interf}}\left(n\right)$ are shown in blue, and the signals from the terahertz receiver $v_{\mathrm{THz}}\left(n\right)$ are shown in red. While the delay range of 330 ps of the ODU includes 14 periods of the signal, for demonstration purposes we will only consider 2 periods here. Because the oscilloscope is triggered on the signal from the interferometer, the traces are already roughly synchronized. To improve the synchronization accuracy, we calculate the envelope $\mathrm{env}\left\{v_{\mathrm{Interf}}\right\}$ shown in yellow, and locate its peaks as indicated by the purple crosses.

We crop the signals from the terahertz receiver $v_{\mathrm{THz}}\left(n\right)$ at the locations of these peaks. This has three benefits:The signals are accurately synchronized.The signals contain exactly one period, thus facilitating a subsequent fast Fourier transform (FFT).The delay axis can be scaled according to the known repetition rate of the MLLD.

After scaling of the delay axis, we can determine that the effective sampling period is 0.23 fs. We perform averaging of the synchronized signals of 100 subsequent measurements without and with the sample, respectively. Given the gain of the transimpedance amplifier, we can scale the amplitude axis. The averaged and correctly scaled traces are depicted in Figure 7. Comparing the signal values at τ=0 and τ=T≈23ps, the periodicity of the signals can be clearly observed. Furthermore, it is evident—as expected—that the introduction of the sample leads to a delay and slight distortion of the signal shape. The delay is due to the sample’s refractive index being greater than that of air, whereas the slight distortion of the signal shape can be attributed to the possibly slightly dispersive nature of the sample and multiple reflections at the interfaces of the sample. To isolate these effects, we further analyze the measurement data in the frequency domain.

#### 3.2.2. Normalization in the Frequency Domain

We find the discrete terahertz spectra by FFT of the time-domain signals shown in Figure 7. The magnitude and phase of the complex spectra within the frequency range from 0 to 1500 GHz are depicted in Figure 8a,b, respectively. Because a single period of the periodic time-domain signal is transformed, the sampling period in the frequency domain exactly matches the frequency resolution F≈43.45GHz of the system. It can be seen in Figure 8a that the measured signal intersects the noise floor at a frequency of 900 GHz. It should be noted that the phase values for frequencies above 900 GHz are not meaningful. Below 900 GHz, we can observe a clear difference in the phase slope without and with the dielectric sample.

Having established a stable phase relationship between the reference measurement and the measurement with the sample in the terahertz path, we can determine the transfer function of the sample relative to a layer of air of the same thickness:(12)$$\begin{array}{c} H_{\mathrm{sample}}\left(f\right)=\frac{{\left.I_{\mathrm{THz},\mathrm{avg}}\left(f\right)\right|}_{\mathrm{Sample}}}{{\left.I_{\mathrm{THz},\mathrm{avg}}\left(f\right)\right|}_{\mathrm{Reference}}}\;. \end{array}$$

Magnitude and phase of the transfer function are depicted in Figure 9a,b, respectively. Since COC exhibits an extremely low loss coefficient, the signal experiences virtually no attenuation as it passes through the 1 mm thick sample. Thus, the magnitude of the transfer function fluctuates around 1. The phase shows a slope of approximately $-0.011\;\frac{\mathrm{rad}}{\mathrm{GHz}}$. Given the sample thickness of 1 mm, we find an average refractive index of 1.517, which matches the value measured with a commercial fiber laser-based THz-TDS system (*Menlo Systems TERA K15*) to the second decimal. We window the transfer function with a 1 THz wide Tukey window with a cosine fraction of 0.5. By applying the inverse FFT after windowing, we obtain the impulse response of the sample depicted in Figure 10. The impulse response exhibits a bandwidth-limited pulse width (full width at half maximum, FWHM) of approximately 0.75 ps and is delayed by 1.74 ps. The bandwidth limitation is mainly due to the finite bandwidth of the terahertz system and the window that has been applied in the frequency domain. It is not due to the frequency characteristics of the dielectric sample.

#### 3.2.3. Results for Thin Samples

Having established the signal processing steps, we can test the phase sensitivity of the system by characterizing thin dielectric samples. To construct dielectric samples with thicknesses that increase linearly in small discrete steps, we stack layers of low-density polyethylene (LDPE) cling film. Each layer has a thickness of d≈17μm as measured with an outside micrometer. Due to their self-adhesive property, the layers of cling film are in close contact and have minimal air inclusions. Considering a refractive index of n≈1.52 [25,26], each layer of cling film is expected to increase the signal delay by
(13)$$\begin{array}{c} \Delta t=\frac{(n-1)\cdot d}{c_{0}}\approx 29.5\;\mathrm{fs}. \end{array}$$

The magnitude and phase of the resulting transfer functions after normalization against a measurement without any layers of cling film are depicted in Figure 11a,b, respectively. There are small yet clearly noticeable undulations both in the magnitude and phase of the transfer function. Most prominently, there is a dip around 650 GHz that cannot be attributed to characteristics of the sample. We have found that these fluctuations in the transfer function are primarily due to errors of the ODU that are not corrected by the approach presented in this paper. These include fluctuations in the speed of the ODU and variations of the optical power with the optical delay.

The associated impulse response is shown in Figure 12. The magnitude of the impulse response is normalized in order to highlight the horizontal shifts of the curves. Because the amount of delay introduced by the cling film is extremely small, the pulse maximum is close to τ=0, and subsequently the left half of the pulse is visible at τ=T.

A magnified view of the pulse maxima is shown as an inset in Figure 12. It can be seen that the measured signal delay increases almost linearly with the number of layers of cling film. On average, each layer adds a delay of approximately 31.5 fs, which matches the calculated value remarkably well.

## 4. Conclusions

In this work, we have proposed and demonstrated an interferometric synchronization method for UHRR-THz-TDS. This method requires only three optical couplers and a pair of low-cost photodiodes for time synchronization with femtosecond accuracy. We have presented the system-theoretical background and explained signal processing steps that are tailored to the ultra-high repetition rate of the system. Finally, we have demonstrated the system’s <30 fs time accuracy that allows us to measure layer thicknesses as thin as 17μm in transmission mode. By adding the interferometer to the UHRR-THz-TDS system, it is capable of highly phase-sensitive applications independent of the ODU’s delay accuracy. This enables the use of simple and low-cost ODUs. In future works, we aim to further improve the performance of the system by using the interferometer signal to correct the delay axis sample by sample.

## Figures and Tables

**Figure 1 sensors-21-05389-f001:**
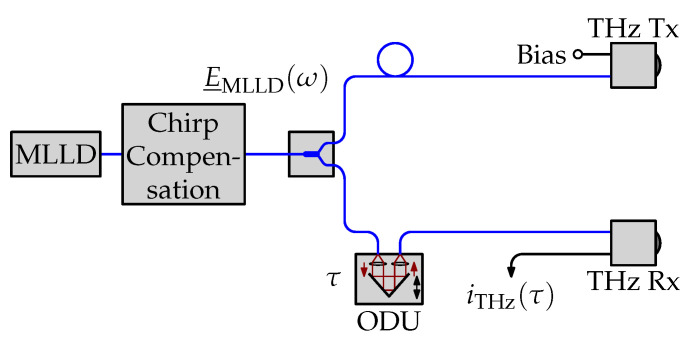
Simplified block diagram of a UHRR-THz-TDS system. Blue lines indicate polarization maintaining fibers. Black lines indicate electrical connections.

**Figure 2 sensors-21-05389-f002:**
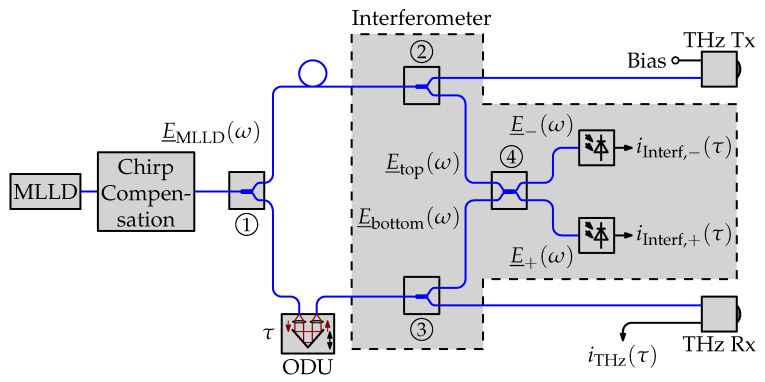
Simplified block diagram of a UHRR-THz-TDS system with interferometer for time synchronization. Blue lines indicate polarization maintaining fibers. Black lines indicate electrical connections. Note the numbering of the optical couplers as used in the text.

**Figure 3 sensors-21-05389-f003:**
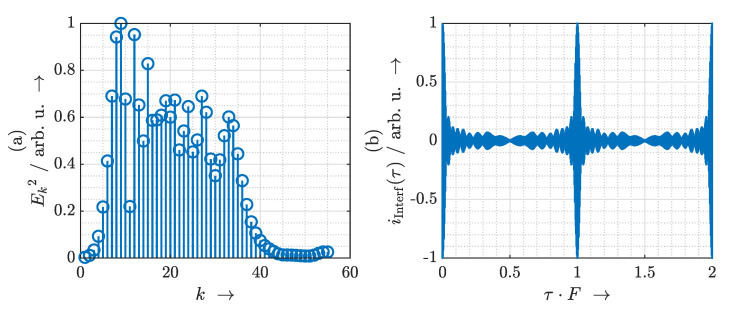
(**a**) Example of an optical spectrum. (**b**) Calculated interferometer signal for the example spectrum.

**Figure 4 sensors-21-05389-f004:**
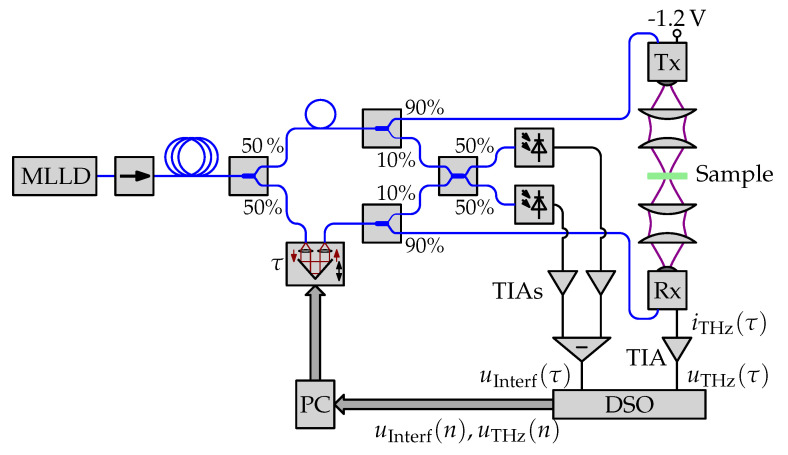
Block diagram of the measurement setup. Blue lines indicate PM fibers. Black lines indicate electrical connections. Bold arrows indicate electrical buses. The terahertz beam path is schematically indicated by purple lines.

**Figure 5 sensors-21-05389-f005:**
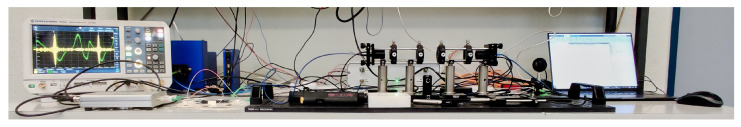
Photograph of the measurement setup.

**Figure 6 sensors-21-05389-f006:**
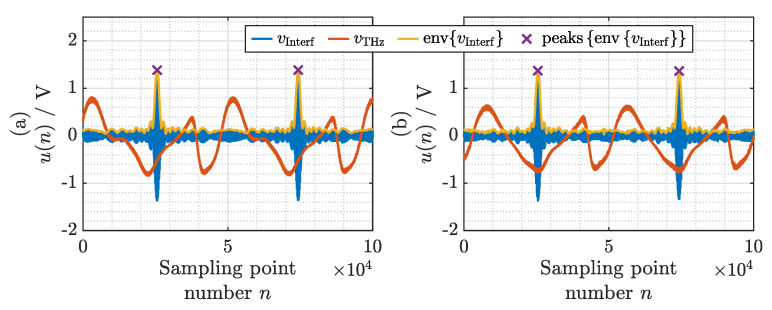
Signals measured with the oscilloscope (**a**) without and (**b**) with the COC sample. Blue lines: signals from the interferometer. Red lines: signals from the terahertz receiver. Yellow lines: envelope of the signal from the interferometer. Purple crosses: maxima of the envelope of the signal from the interferometer.

**Figure 7 sensors-21-05389-f007:**
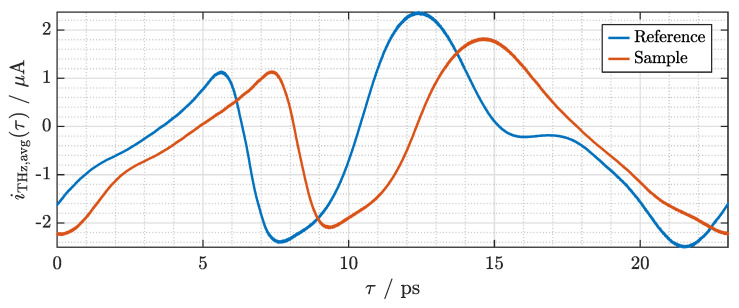
Photocurrents from the terahertz receiver after cropping, 100-times averaging, and scaling of the delay and amplitude axes. Blue: without the sample. Red: with the COC sample.

**Figure 8 sensors-21-05389-f008:**
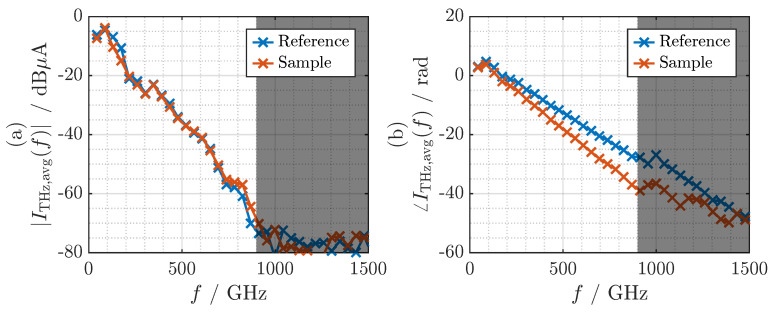
(**a**) Magnitude and (**b**) phase of the Fourier transform of the photocurrents from the terahertz receiver. Blue: without the sample. Red: with the COC sample.

**Figure 9 sensors-21-05389-f009:**
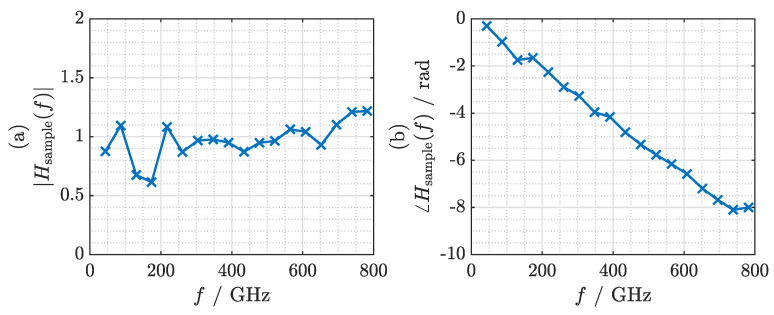
(**a**) Magnitude and (**b**) phase of the transfer function of the COC sample.

**Figure 10 sensors-21-05389-f010:**
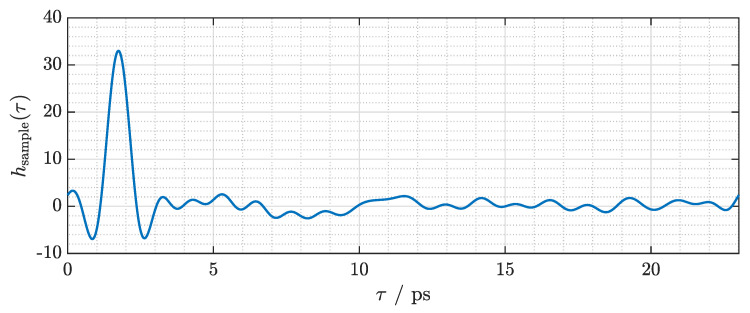
Impulse response of the COC sample.

**Figure 11 sensors-21-05389-f011:**
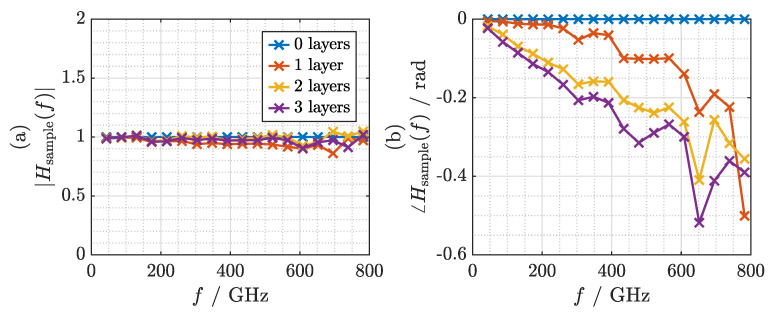
Processed measurement results for 0–3 layers of cling film. (**a**) Magnitude of the transfer function. (**b**) Phase of the transfer function.

**Figure 12 sensors-21-05389-f012:**
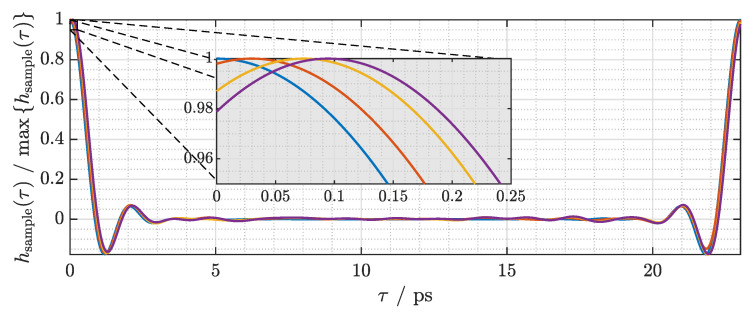
Impulse response of 0–3 layers of cling film.

## Data Availability

The data presented in this study are available on request from the corresponding author.

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
