# Peer review of "Ultra-High Repetition Rate Terahertz Time-Domain Spectroscopy for Micrometer Layer Thickness Measurement"

_sensors, 2021, doi:10.3390/s21165389_

Round 1

Reviewer 1 Report

This paper presents a terahertz-time domain spectroscopy system with ultra-high repetition rate, which could be used to monitor micrometer layer thickness. It is an interesting work on thin layer measurements. The reviewer has the following comments.

  1. A picture of the setup should be preferred.
  2. In lines 216-217, the authors demonstrated that the introduction of the sample leads to a slight distortion of the signal shape. They should explain why and how they isolate its influence.
  3. There is an obvious dip at 700 GHz in Fig. 10(b), the authors should specify that and explain why.
  4. The structure of the manuscript is expected to be improved for better understanding.

Reviewer 2 Report

In the section starting line 63 you describe your method, as if it would replace 1:1 that of ref. 22 and is even more less complex. Please correct this and point out, that ref. 22 uses fs-scale timing accuracy for every sample of the terahertz waveform that is acquired, whereas you use your optical reference to "pin" the waveform itself to a reference. I realized, that you mention in the conclusions (last sentence), that you want to approach ref. 22 as a next step, but be more precise in the introduction and in the main content of the paper. 

Concerning the thickness measurements: You claim to be able to measure very thin samples, which you also demonstrate. But you know, that in terahertz applications, the measurement of sample thicknesses is mostly considered in reflection. So, be fair to directly point out in every point you mention this, that you are measuring in transmission. Transmission measurements of thin samples are much less demanding than reflection measurements. 

Nevertheless, congratulations on this nice work and progress in the field of your UHHR setups. Given that you change the points mentioned above, I then recommend the publication of your manuscript. 

suggestions line 3: fiber-laser-driven systems
Abstract: Add the information, that the stacked layers are measured in transmission. 
line 68/69: Add the information, that the stacked layers are measured in transmission. 
Conclusions line 270: Add the information, that the samples are measured in transmission. 
